

**Biotic factors dominantly determine soil inorganic carbon stock across**
**Tibetan alpine grasslands**
Junxiao Pan [a], Jinsong Wang [a,*], Dashuan Tian [a], Ruiyang Zhang [a], Yang Li
[a], Lei Song [a,b], Jiaming Yang [a], Chunxue Wei [a], Shuli Niu [a,b,*]
[a] Key Laboratory of Ecosystem Network Observation and Modeling, Institute of
Geographic Sciences and Natural Resources Research, Chinese Academy of Sciences,
Beijing 100101, PR China
[b] College of Resources and Environment, University of Chinese Academy of Sciences,
Beijing 100049, PR China
*Corresponding author at: Key Laboratory of Ecosystem Network Observation and
Modeling, Institute of Geographic Sciences and Natural Resources Research, Chinese
Academy of Sciences, Beijing 100101, PR China.
E-mail address: wangjinsong@igsnrr.ac.cn (J. Wang), sniu@igsnrr.ac.cn (S. Niu).





**Abstract.** Soil inorganic carbon (SIC) pool is a major component of soil C pools, and
clarifying the predictors of SIC stock is urgent for decreasing soil C losses and
maintaining soil health and ecosystem functions. However, the drivers and their relative
effects on the SIC stock at different soil depths remain largely unexplored. Here, we
conducted a large-scale sampling to investigate the effects and relative contributions of
abiotic (climate and soil) and biotic (plant and microbe) drivers on the SIC stock
between topsoils (0–10 cm) and subsoils (20–30 cm) across Tibetan alpine grasslands.
Results showed that the SIC stock had no significant differences between the topsoil
and subsoil. The SIC stock was positively associated with altitude, pH, and sand
proportion, but negatively correlated with mean annual precipitation, plant
aboveground biomass, plant coverage, root biomass, soil available nitrogen, microbial
biomass carbon, and bacterial and fungal gene abundance. For both soil layers, biotic
factors had larger effects on the SIC stock than abiotic factors did. But the relative
importance of these determinants varied with soil depth, with the effects of plant and
microbial variables on SIC stock weakening with soil depth, whereas the importance of
climatic and edaphic variables increasing with soil depth. Specifically, bacterial and
fungal gene abundance and plant coverage played dominant roles in regulating SIC
stock in the topsoil, while soil pH contributed largely to the variation of SIC stock in
the subsoil. Our findings highlight differential drivers over SIC stock with soil depth,
which should be considered in biogeochemical models for better simulating and
predicting SIC dynamics and its feedbacks to environmental changes.



## 1 Introduction

Soils store approximately 1,500 Pg of organic carbon (SOC) and 940 Pg of inorganic carbon (SIC) to a depth of 1 m (Batjes, 1996; Jobbágy & Jackson, 2000), which are the largest carbon (C) pool in the terrestrial ecosystem and play a critical part in the global C cycling (Darwish et al., 2018; Lal 2004; Prietzel et al, 2016). Compared to the relatively short turnover time of SOC, SIC has a long residence time due to soil weathering (Monger et al, 2015; Zang et al, 2018), which is considered to be fairly stable and has less contribution to changes in terrestrial ecosystem C balance (Yang et al, 2012). Therefore, previous studies have paid little attention to SIC. However, recent studies suggest that SIC is also responsive to anthropogenic activities and global climate changes such as soil acidification, atmospheric N deposition, and global warming (Yang et al, 2010; Song et al, 2022), acting as a critical C source (Liu et al, 2020) or C sink (Gao et al, 2018; Liu et al, 2021). Thus, the preservation of SIC and its roles in climate mitigation should not be neglected, especially in arid and semi-arid grasslands where store a large amount of SIC (Yang et al, 2012).

SIC stock and stability can be fundamentally altered by an array of abiotic and biotic processes (Raza et al, 2020). High precipitation can promote soil silicate minerals weathering and removal of base cations ($Ca^{2+}$, $Mg^{2+}$, $K^+$, and $Na^+$) by leaching (Vicca et al 2022). Soil acidification due to atmospheric nitrogen (N) and acid deposition and the nitrification of $NH_4^+$ may greatly accelerate soil carbonate dissolution and $CO_2$ releases (Raza et al, 2020; Song et al, 2022). Plant growth can deplete soil carbonates by releasing proton and organic acids from root rhizosphere (Goulding et al, 2016;

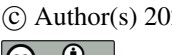



Kuzyakov & Razavi, 2019), and biological $N_2$ fixation by some legumes are likely to
cause SIC losses (Tang et al, 1999). Furthermore, plant autotrophic and microbial
heterotrophic respiration often facilitate carbonate dissolution by enhancing $CO_2$ partial
pressures (An et al, 2019; Liu et al, 2021). Nevertheless, how these abiotic and biotic
factors affect SIC stock and what is the relative importance of these confounding drivers
remain largely uncertain.
Previous studies on SIC stock mostly have focused on the topsoil, while the patterns
of SIC stock in the subsoil on a large scale remain elusive. The predictors of SIC stock
in the subsoil may differ from those in the topsoil due to distinct soil microenvironments,
soil physicochemical properties, root exudates, and microbial abundance and functions
(Jia et al, 2017). For instance, the topsoil has larger root biomass and higher microbial
activity than the subsoil, but the subsoil tends to preserve soil parent material because
of the weakened weathering by the isolation of heat and energy from the surface soil
(Crowther et al, 2016). Thus, the abiotic and biotic variables may exhibit different
effects on SIC stock in the subsoil compared to the topsoil due to the various importance
of these variables.
The Tibetan Plateau has the largest alpine grassland on the Eurasian continent,
which is a vital component of global terrestrial ecosystems, providing an ideal platform
to explore SIC stock and its determinants (Wang et al, 2002; Yang et al, 2010). During
the past several decades, the plateau has experienced significant warming (Wang et al,
2008) and pronounced atmospheric N deposition (Liu et al, 2013; Yu et al, 2019). This
continuous warming and N deposition have resulted in a significant increase in plant



growth and soil acidification (Ding et al, 2017; Yang et al, 2012), which could be likely
to induce potential $CO_2$ releases from soil carbonates by biogeochemical process (Raza
et al, 2020). However, a general understanding of SIC stock with soil depth across
Tibetan alpine grasslands remains unexplored. Here, we researched the relative
importance of climatic, edaphic, plant, and microbial variables to SIC stock at different
soil layers along an approximately 3,000 km transect of alpine grasslands on the Tibetan
Plateau, spanning a broad range of climatic and geographical conditions. Specifically,
two key questions are addressed in this study: (1) what are the differences of SIC stock
between the topsoil and subsoil? (2) how does the relative importance of climatic,
edaphic, plant, and microbial variables to the variation of SIC stock along with soil
depth?



## 2 Material and methods

### 2.1 Study area and field sampling

During July, August, and September 2020, we conducted large-scale systematic field surveys and samplings in Tibetan alpine grasslands. The total 25 sampling sites covered approximately 3,000 km and included three grassland types (i.e, 11 alpine meadow, 8 alpine steppe, and 6 alpine desert sites). The distance between nearby sampling sites was about 120 km. The study sites cover a broad geographic and climatic range, with longitude and latitude ranging from 79°49'39" to 102°25'31" E and 31°06'37" to 32°43'09" N, respectively, and the altitude ranging from 3500 m to 5016 m. These sites covered a broad precipitation gradient varying between 72 mm and 706 mm. The mean annual temperature (MAT) ranged from –3.9°C to 5.8°C. The plant communities were dominated by *Kobresia tibetica Maxim*, *Stipa caucasica*, *Kobresia pygmaea*, *Stipa purpurea*, and *Leontopodium pusillum*. Soils were *Cambisol* and some were loess-derived *Luvisol*. The site location, grassland type, climatic, and plant parameters were detailed in Table S1.

### 2.2 Climatic data

The climatic data were derived from the LPSDC (Loess Plateau Scientific Data Center, http://loess.geodata.cn/) (Peng et al, 2019). The Kriging interpolation was conducted to obtain spatial distributions of 30-year MAT and MAP (1987-2017) at each sampling site by a geographic coordinate system.



**2.3 Soil properties**
At each site, we selected four 1 m ×1 m plots for soil and plant samplings and the
distance between nearby sampling plots was 25 m. In each plot, a 7.5-cm diameter soil
drill was used to take five soil cores at fixed soil depths (0-10 cm, 10-20 cm, and 20-30
cm), and a 2-mm mesh was used to remove stones. We used soil samples from 0–10 cm
and 20–30 cm to represent the topsoil and subsoil, respectively, according to previous
studies (Angst et al, 2021; Rumpel & Kögel-Knabner 2011; Zhou et al., 2021). After
mixing, 100 g of fresh soils from each plot were collected and stored in a –4°C portable
icebox, then returned to the laboratory and stored at –20°C for microbial properties.
The rest soil samples about 700 g were also sent back to the laboratory and air-dried for
measurements of other soil properties. A 40 cm ×40 cm ×40 cm (length × width × depth)
pit was dug for measuring soil bulk density (BD) by using a constant volume soil
sampling drill (100 cm$^3$), and the undisturbed soil was preserved in aluminum specimen
boxes returning to the laboratory and oven-dried for 48 hours at 105°C and weighed.
The oven-dried soil (20 g) was screened into gravel by sifting through a 2-mm mesh
sieve and gravels larger than 2 mm were collected and weighed to determine the
percentage of gravels. Soil pH (1:25 soil: H$_2$O) was measured using a soil pH meter,
and available nitrogen (AN) was determined by the alkaline-hydrolysis diffusion
method. A laser particle analyzer (Mastersizer 2000, Malvern Panalytical, UK) was
applied to measure soil mechanical compositions, including clay (< 2 μm), silt (2-50
μm), and sand (> 50 μm) proportion. SIC was determined by using an inorganic C
analyzer (multi EA® 4000; Analytic Jena, Germany). The multi EA 4000 C elemental



analyzer was equipped with the automatic TIC solids module and calibrated before the
analysis. The sample boat was acidified automatically with 40 % $H_3PO_4$ in the reactor
of the TIC module. And the $CO_2$ from the carbonate was released, the measuring gas
was dried and cleaned and the carbon content was measured by means of the wide-
range NDIR detector. Before being analyzed directly, all soil samples were ground into
solid fine powders with a mortar, and for the determination of TIC, a standard, prepared
by solids-dilution of $CaCO_3$ with $SiO_2$ (0.2 % C), was used, with weighting rage 7-200
mg, to cover a wide concentration range.
**2.4 Plant properties**
In each plot, we estimated plant coverage (PC) by the projection method, namely the
proportion of vegetation projection to the area of the sampling plot. In addition, plant
aboveground biomass and belowground roots were clipped and collected, respectively,
then oven-dried at 60°C and weighed to determine plant aboveground biomass (PAB)
and root biomass (RB).
**2.5 Microbial attributes**
Soil microbial biomass carbon (MBC) was measured by using a chloroform
fumigation-extraction procedure (Brookes et al, 1985). Briefly, 10 g of unfumigated
and chloroform-fumigated fresh soil samples were extracted by using 0.5 M $K_2SO_4$
after 24 h of incubation, respectively. Then, the extracts were analyzed by using a TOC
analyzer (multi N/C® 3100; Analytic Jena, Germany). The MBC was determined by



the differences in C concentrations between unfumigated and chloroform-fumigated
samples, and the correction factor (i.e, KC= 0.45) was used to convert microbial C to
MBC (Joergensen, 1996).

Real-time polymerase chain reaction (qPCR) was used to quantify bacterial (BA)

and fungal gene abundance (FA) by the absolute quantification method based on the
gene copy number (Tatti et al, 2016). Each reaction was carried out 3 times with a
mixture of a total 20 µL volume, including 2 µL of DNA template, 10 µL of 2× ChamQ
SYBR Color qPCR Master Mix, and 0.4 µL (5µM concentration) each of forward and
reverse primer specific for each gene. And the PCR conditions were 95℃ for 5 min,
then 40 cycles for the 18S rRNA gene and 16S rRNA gene. Each cycle involved melting
at 95℃ for 30 s, annealing at 55℃ for 30 s, an extension of 72℃ for 40 s, and finally
10℃ until terminated. And the primer pair SSU0817/1196 and Eub338/Eub806 were
used for amplifying fungi and bacteria in PCR amplification, respectively. Then the
DNA concentration was determined by using a QuantiFluor™-ST fluorescent
quantitative system (Promega, Fitchburg, WI, USA). The abbreviations of all variables
were detailed in Table S2.
**2.6 Statistical analyses**
The total SIC density (C stock per land area) in each soil depth layer was calculated
using Equation (1) (Pan et al, 2019):
SIC density (g C m$^{-2}$) = SIC (g C kg$^{-1}$) × BD (g cm$^{-3}$) ×d (cm) × (1-g) /100          (1)



where SIC is soil inorganic C content, d is the depth of the soil layer (0.1 m), BD is
bulk density, and g is the percentage of gravel fraction (>2 mm).

First, the differences of SIC stock and corresponding abiotic and biotic variables

between the topsoil and subsoil were examined by $T$-test. Second, SIC density and
various abiotic and biotic variables were log-transformed and standardized (z-score
normalization) to perform the assumption of normality and homogeneity by Shapiro-
Wilk and Levene's test, respectively (Pan et al, 2021). Then the linear regressions were
used to test SIC density about different variables for both the topsoil and subsoil across
sites.

Third, a linear model was employed to examine SIC density with abiotic and biotic

variables by using the maximum likelihood estimation with the lm package. And the
relative effect of the parameter estimates was calculated to evaluate the relative
importance of drivers controlling SIC density. Also, SIC density and abiotic and biotic
variables were standardized before analyses, using the Z-score to interpret variable
estimates on a comparable scale (Gross et al, 2017).
Log (SIC density) = $\beta_0 + \beta_1 \log X_1 + \beta_1 \log X_2 + \ldots \beta_{12} \log X_{12}$                    (2)

where $\beta_0$ and $\beta_i$ (i=1, 2, 3…12) are intercept and coefficients, respectively. To

explore the determinants of SIC density in different soil depths across all sites, the
absolute values of slopes of the variables were extracted and plotted. Then, 12
controlling variables were categorized into four groups, including climatic (MAP, MAT,
and altitude), edaphic (pH, AN, and sand proportion), plant (PB, PC, and RB), and



microbial (MBC, BA, and FA) factors, to quantify their relative contribution to SIC
density (Fang et al, 2019).

Furthermore, the relative importance of abiotic (climatic and edaphic) and biotic

(plant and microbial) variables in determining SIC density was quantified by
performing variation partitioning analyses (VPA) by using the "vegan" package in R

4.1.3.






## 3 Results

### 3.1 SIC density and influencing variables in different soil depths

SIC density and SIC content had no significant differences between the topsoil and subsoil, but bulk density in the subsoil was much higher compared with the topsoil. Specifically, SIC density in the topsoil and subsoil ranged from 1.8 g C m$^{-2}$ to 3271 g C m$^{-2}$ and 5.4 g C m$^{-2}$ to 3214 g C m$^{-2}$ across 25 sampling sites, with an average of 802 ± 220 g C m$^{-2}$ and 814 ± 236 g C m$^{-2}$, respectively (Fig. 1). No significant changes in SIC density with soil depth were observed in both the alpine steppe and alpine desert ($p$=0.113 and $p$=0.068, respectively; Fig. 1), but SIC density was higher in the subsoil than that in the topsoil in the alpine meadow ($p = 0.002$, Fig. 1).

Meanwhile, the majority of abiotic and biotic drivers had significant differences between the topsoil and subsoil (Table 1). RB, AN, MBC, BA, and FA in the topsoil were significantly larger than those in the subsoil (all $p<0.001$). In contrast, pH was significantly lower in the topsoil than in the subsoil ($p<0.001$, Table 1). However, the sand proportion between the two soil depths had no significant differences (Table 1).

### 3.2 Associations of SIC density with abiotic and biotic variables

The SIC density was closely related to multiple abiotic and biotic variables (Fig.s 2 and 3). For both the topsoil and subsoil, the SIC density was positively associated with altitude, pH, and sand proportion, but negatively correlated with MAP, PAB, PC, RB, AN, BA, and FA. The SIC density showed a negative correlation with MBC in the

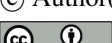



topsoil (Fig. 2), but not in the subsoil (Fig. 3). Meanwhile, the SIC density in both two
soil depths did not correlate with MAT (Figs. 2 and 3).
**3.3 Determinants of SIC density in different soil depths**
The linear model and VPA collectively displayed that the predominant drivers of SIC
density differed with soil depth (Figs. 4 and 5). Specifically, for the topsoil, the linear
model revealed that microbial and plant variables largely explained the variations in the
SIC density, followed by edaphic variables and climate contributed the least (Fig. 4).
Among these variables, PC, BA, and FA exhibited larger effects on the SIC density
compared with other controlling factors (Fig. 4). Also, the VPA analysis illustrated that
biotic factors explained the majority variation of SIC density compared with abiotic
factors (Fig. 5). For the subsoil, the linear model showed that edaphic variables largely
explained the variation in SIC density, followed by microbial and plant variables, and
climate contributed the least (Fig. 4). Among these variables, the soil pH had larger
contributions to the variation of SIC density rather than others (Fig. 4). Meanwhile, the
VPA analysis confirmed that the effects of biotic factors on SIC density were larger
than those of abiotic factors in the subsoil (Fig. 5).



## 4 Discussion


To the best of our knowledge, this study was the first to afford large-scale evidence of
the relative contribution of abiotic and biotic drivers to the variation of SIC stock at
different soil depths, which has considerable implications for grasping the importance
of SIC in the ecosystem C cycling. Since considerably stable characteristics and the
long turnover time (Mi et al, 2008; Yang et al, 2010; Zamanian et al, 2018), SIC stock
is traditionally considered to be dominated by abiotic factors including soil moisture,
soil pH, $CO_2$ partial pressure, and $Ca^{2+}$ concentrations according to the equilibrium of
carbonate precipitation–dissolution reactions ($CaCO_3 + H_2O + CO_2 \rightarrow Ca^{2+} + 2HCO_3^-$
and $Ca^{2+} + 2HCO_3^- \rightarrow CaCO_3 + H_2O + CO_2$) and mineral carbonation ($MgSiO_4 + 2CO_2$
$\rightarrow 2MgCO_3 + SiO_2$ and $CaMgSi_2O_6 + CO_2 + H_2O \rightarrow Ca_2Mg_5Si_8O_{22}(OH)_2 + CaCO_3 +$
$SiO_2$) (Mi et al, 2008; Rey, 2015; Yang et al, 2012; Yang and Yang, 2020). These abiotic
factors were proved to have large impacts on the dissolution and deposition processes
of inorganic C and ultimately determined the reservation and distribution of SIC (Rey,
2015; Rowley et al, 2018).
However, many biological processes and factors were not quantitatively considered
in previous studies. In this study, based on the approach of large-scale field samplings
across Tibetan alpine grasslands, we estimated the predominant drivers of SIC stock in
the topsoil and subsoil. Our results found the predominant roles of microbial and plant
factors in determining SIC stock in both topsoil and subsoil. More importantly, the
effects of biotic factors on SIC stock weakened with soil depth (Fig. 4). These results
were different from those demonstrating the critical influence of abiotic processes on



SIC stock (Mi et al, 2008; Yang et al, 2010).
We found that increasing plant aboveground biomass, plant coverage, and root
biomass significantly decreased SIC density (Figs. 2 and 3). Plant factors could
contribute to the decline of SIC stock by three pathways including uptakes of
exchangeable cations, plant organic matter inputs, and rhizosphere processes. First, a
large decline in soil base cations is likely to be induced by plant uptake with increasing
plant biomass. And the losses of soil exchangeable base cations can cause the
transformation of SIC to $CO_2$, which is ultimately released into the atmosphere (Huang
et al, 2015). Second, increasing plant residue inputs can enhance carbonic and organic
acid production into soil water solution via microbial decomposition, which reduces the
availability of soil base cations through cation exchange in the soil (Sartori et al, 2007)
and increase the dissolution and leaching of carbonates, resulting in a decrease in the
SIC. Third, the plant rhizosphere effect on releasing $CO_2$ from carbonates should not
be ignored, especially in alkaline soils. By releasing organic acids and protons as well
as $CO_2$, plant roots can reduce soil pH and increase $CO_2$ in the rhizosphere (Lenzewski
et al, 2018), both of which dissolve carbonates by neutralization (Harley & Gilkes,
2000). In addition, organic compounds from plant root exudates, such as malate or
citrate, can stimulate mineral weathering by dissolving silicate minerals (Dontsova et
al, 2020).
Furthermore, the topsoil has a larger quantity and higher quality of plant residues
than the subsoil, which indicates a more potential for carbonate dissolution by
biological processes for the surface soil (Liu et al, 2020). The large root biomass in the



topsoil can increase the uptake of base cations and result in increasing proton and
organic acids in root exudates (Li et al, 2007), thus reducing the soil carbonate content
for maintaining the charge balance. In addition, the larger plant roots exuded more
organic compounds in the topsoil that can stimulate parent mineral weathering and
dissolve silicate minerals by chelating reaction products (Doetterl et al, 2015; Dontsova
et al, 2020).
Previous studies reported that microbial properties may not be important in
mediating SIC accumulation (Liu et al, 2021; Wang et al, 2015). However, our results
found that microbial factors including microbial biomass and bacterial and fungal gene
abundance showed significant and negative associations with SIC stock (Figs. 2 and 3),
which could be due to microbes driving the carbonate dissolution processes, including
microbial respiration, organic matter mineralization, and releases of proton and organic
acids by microbial metabolic activity. First, the increase in microbial respiration can
improve $CO_2$ production and enhance the partial pressure of $CO_2$, leading to a decline
in pH and further dissolution of carbonates (Chang et al, 2012). In addition, soil organic
matter mineralization and litter decomposition by microbes can induce the dissolution
of $CO_2$ and the release of organic acids (Goulding, 2016; Kuzyakov & Razavi, 2019),
both of which decrease the SIC stock. Meanwhile, chelates and enzymes excreted by
microbes may contribute to enhancing mineral dissolution rates and organic matter
decomposition (Xiao et al, 2015; Zaharescu et al, 2020).
We also revealed that bacterial and fungal gene abundance contributed significantly
to the variation of SIC stock (Figs. 2 and 3), which was likely to account for decreasing





soil pH in the involvement of microbial biological reactions. For instance, nitrifying
bacteria can oxidize ammonium to nitrate ($NH_4^+ + OH^- + 2O_2 \rightarrow NO_3^- + 2H_2O + H^+$),
and the production of acidity is finally neutralized through accelerating carbonate
dissolution (Zamanian et al, 2016). Also, some nitrogen-fixing bacteria that lived in
symbiosis with leguminous plants can acidify the soil by excreting protons during $N_2$
fixation (Vicca et al, 2022). Furthermore, fungi are likely to accelerate carbonate
neutralization by exuding protons and organic acids (Van Hees et al, 2006; Wild et al,

2021).

Microbial factors also affected SIC stock more in the topsoil than in the subsoil.
The large plant residues incorporated into the topsoil provided substantial amounts of
organic matter for microbial living and decomposition (Oelkers et al, 2015; Ven et al,
2020), which can stimulate microbial abundance and activities and promote microbial
extracellular enzymes. These extracellular excretions play a fundamental role in
microbial respiration and $CO_2$ production, both of which stimulate silicate weathering
and carbonate dissolution (Vicca et al, 2022). Meanwhile, the higher $CO_2$ flux and $CO_2$
partial pressure resulting from the biological activities of roots and soil microorganisms
in the topsoil could enhance carbonate dissolution and formations of pedogenic
inorganic C (Chang et al, 2012; Zamanian et al, 2016).
Different from plant and microbial factors, the effects of edaphic factors on SIC
stock strengthened with soil depth, with soil pH being the most important predictor
among edaphic variables (Fig. 4). The buffering capacity in soil solutions determines
the equilibrium of ion inputs and outputs by soil pH (Huang et al, 2015). In this study,



soil pH in the subsoil (7.85) was much higher than that (7.66) in the topsoil (Table 1).
The higher pH could buffer the replacement of the exchangeable cations with protons
(Frank & Stuanes, 2003) and increase the preservation of base cations (Gandois et al,
2011). Given that base cations and carbonates provide the major buffering capacity in
the alkaline soil (Yang et al, 2012), the topsoil could be subject to a larger loss of base
cations and SIC due to the lower soil pH compared to the subsoil.
Taken together, our results revealed that SIC stock was closely linked with biotic
factors, which highlights the roles of biological processes in regulating SIC dynamics
(Hong et al, 2019). These results imply that the widespread enhancement of vegetation
productivity under global environmental changes (e.g, warming and rewetting) (Ding
et al, 2017; Wang et al, 2008) may aggravate the depletion of SIC stock (Raza et al,
2020). Meanwhile, previous studies have urged the need for incorporating microbial
processes and indicators into Earth system models (ESMs) to reduce the uncertainty in
predicting soil C dynamics, especially SOC decomposition (Allison et al, 2010;
Moorhead and Sinsabaugh, 2006; Todd-Brown et al, 2013). However, our findings
highlighted the vital role of microbial factors in regulating soil C balance from
inorganic C preservation. Thus, incorporating microbial processes into the models can
aid in the understanding of overall soil C responses, because SOC and SIC are formed,
protected, and lost in different ways.
More importantly, the effects of biotic factors on SIC stock weakened with soil
depth, which implies that SIC may be susceptive to environmental changes in the
topsoil where is the hotspot of root and microbial activities. Even though biotic factors





in the subsoil played less roles in affecting SIC stock compared with the topsoil, an
increase in rooting depth is expected in response to climate warming and land-use
change (Liu et al. 2018), which is likely to cause SIC losses in the deep soil by root
growth. Therefore, it is a necessity to further explore the effects of biotic factors on SIC
stock in the deep soil in the context of global changes. Overall, the contribution of SIC
to $CO_2$ is not ignored and SIC maintenance has a considerable significance on soil C
losses and maintains the health and ecosystem functions (Raza et al, 2020; Zamanian
et al, 2018). Our study provides robust evidence that biotic factors are mainly
responsible for the variation of SIC stock and that topsoils and subsoils should be
considered separately when modeling SIC dynamics and its feedbacks on climate
change (Yang et al, 2012; Zamanian & Kuzyakov, 2019).

**5 Conclusions**

Our findings showed that the climatic, edaphic, plant, and microbial variables jointly
affected SIC stock in the Tibetan grasslands and that biotic factors had a larger
contribution than abiotic factors to the variation of SIC stock. Furthermore, the effects
of microbial and plant variables on SIC stock weakened with soil depth, while the
effects of edaphic variables strengthened with soil depth. The contrasting responses and
drivers of SIC stock between the topsoil and subsoil highlight differential mechanisms
underlying SIC preservation with soil depth, which is crucial to understanding and
predicting SIC dynamics and its feedbacks to environmental changes.



**Data availability.**

The data that support the findings of this study are available from the corresponding

author upon reasonable request.

**Supplement.**

Supporting information is also available as supplementary material.

**Author contributions.**

JP, JW, and SN designed the study. JP, JW, DT, RZ, YL, LS, JY, CW, and SN were

involved in drafting or revising the manuscript. All authors read and approved the

final manuscript.

**Competing interests.**

The authors declare that they have no conflict of interest.

**Acknowledgments**

This study was financially supported by the Second Tibetan Plateau Scientific

Expedition and Research (STEP) program (2019QZKK0302), the National Natural

Science Foundation of China (31988102 and 32101390), and the China National

Postdoctoral Program for Innovative Talents (BX20200330).



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





## Figure captions


**Figure 1.** Soil inorganic C content, bulk density, and SIC density in the topsoil and
subsoil. The horizontal solid and hollow lines inside each box represent medians and
mean values, respectively. Significant differences between the topsoil and subsoil were
inspected according to Tukey's test.
**Figure 2.** SIC density in relation to climatic, edaphic, plant, and microbial factors in
the topsoil. The solid lines are fitted by ordinary least-squares regressions, and the
shadow areas correspond to 95% confidence intervals. AM: alpine meadow; AS: alpine
steppe; AD: alpine desert; MAP: mean annual precipitation; PAB: plant aboveground
biomass; PC: plant coverage. The abbreviations for other variables are shown in Table
1. *$p<0.05$; **$p<0.01$; ***$p<0.001$.
**Figure 3.** SIC density in relation to climatic, edaphic, plant, and microbial factors in
the subsoil. The solid lines are fitted by ordinary least-squares regressions, and the
shadow areas correspond to 95% confidence intervals. AM: alpine meadow; AS: alpine
steppe; AD: alpine desert.
**Figure 4.** Relative effects of multiple drivers of SIC density in the topsoil (A) and
subsoil(B). Climatic variables include MAP, MAT, and altitude; edaphic variables
include pH, AN, and sand proportion; plant variables include PB, PC, and RB;
microbial variables include MBC, BA, and FA.
**Figure 5.** Variation partitioning analyses (VPA) reveal the relative contribution of
abiotic and biotic variables to SIC density in the (A) topsoil (61.2% vs. 84.4%) and (B)
subsoil (73.4% vs. 86.1%), respectively. Results in three fractions: the unique effect of



abiotic factors (X1), the unique effect of biotic factors (X2), and common interception
of abiotic and biotic factors (X3).





**Figure 1.**

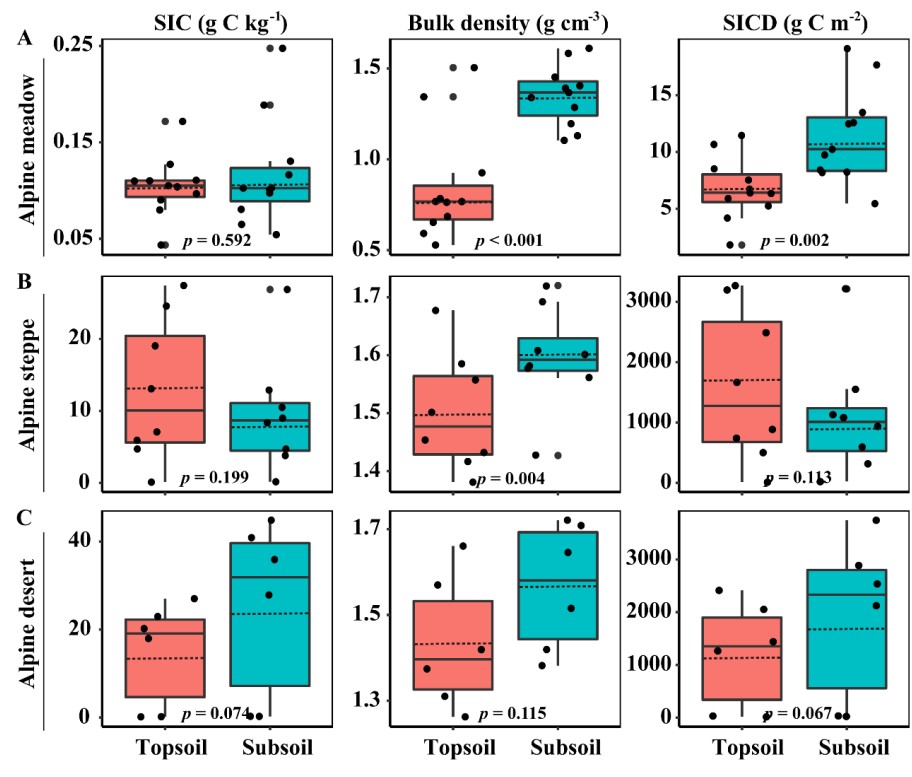







**Figure 2.**

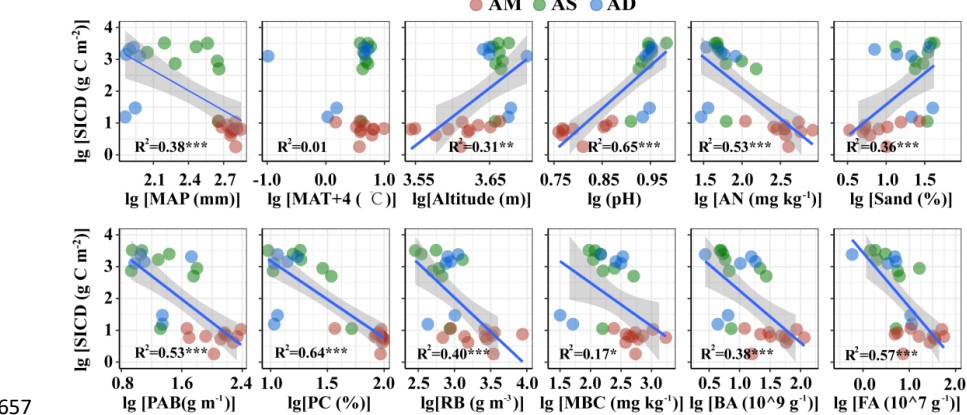






**Figure 3.**

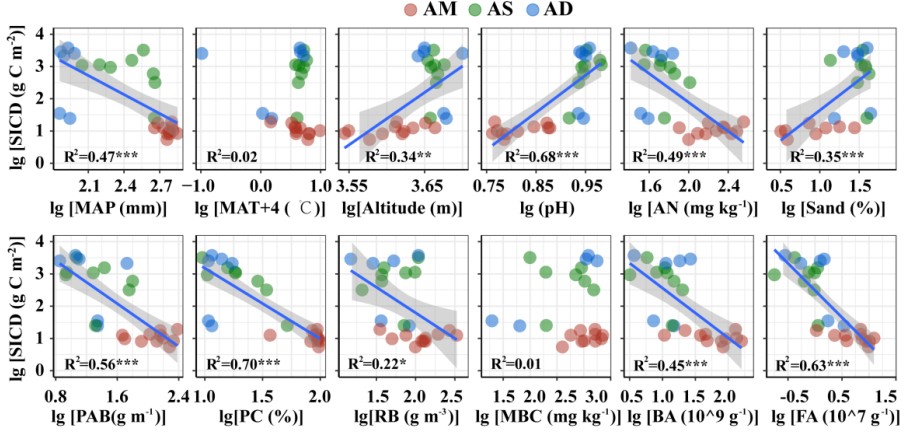




**Figure 4.**

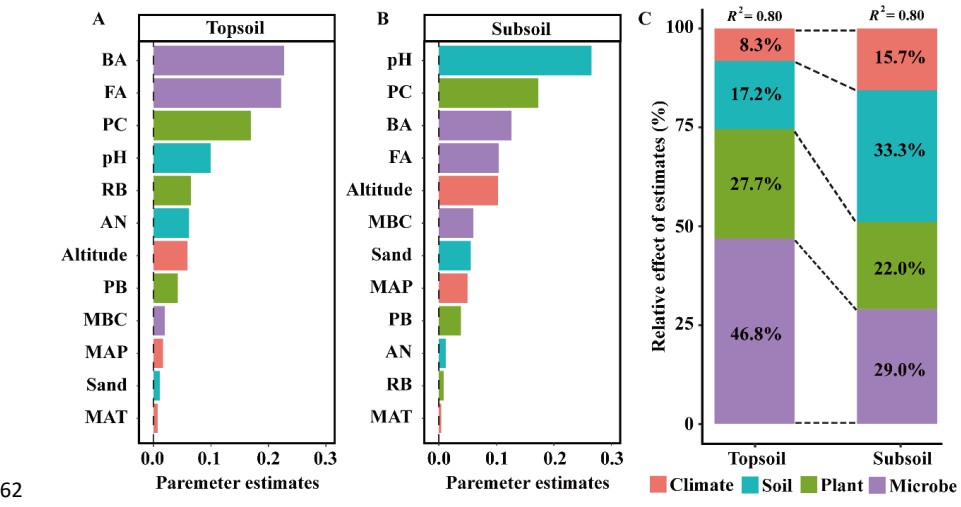



**Figure 5.**

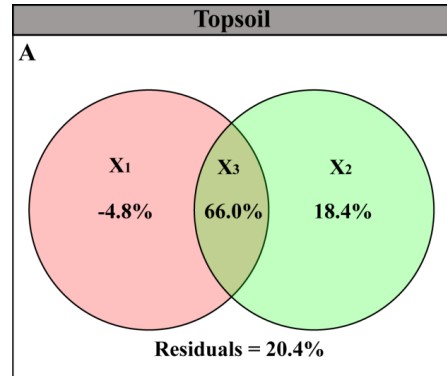
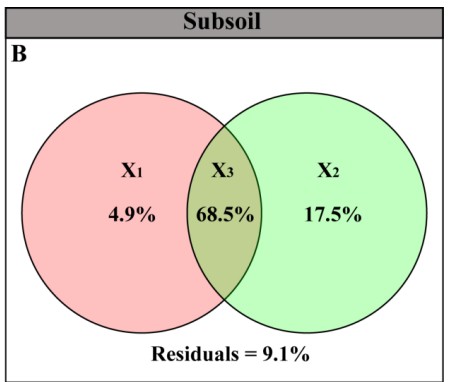



**Table 1.** Edaphic, plant, and microbial properties between the topsoil and subsoil for
25 sampling sites.

| Parameters | Topsoil | Subsoil | *p* value |
|---|---|---|---|
| RB (g m$^{-2}$) | 1670 ±359 | 95.2 ±15.3 | <0.001 |
| pH | 7.66 ±0.28 | 7.85 ±0.26 | <0.001 |
| AN (mg kg$^{-1}$) | 217 ±43.7 | 131 ±22.0 | 0.004 |
| SP (%) | 47.1 ±4.33 | 45.6 ±4.87 | 0.698 |
| MBC (mg kg$^{-1}$) | 385 ±73.8 | 101 ±9.7 | 0.001 |
| BA (10^9 gene copies g$^{-1}$ soil) | 27.2 ±5.68 | 12.6 ±2.86 | 0.001 |
| FA (10^7 gene copies g$^{-1}$ soil) | 14.2 ±3.25 | 3.62 ±0.84 | 0.001 |

RB: root biomass; AN: soil available nitrogen; SP: sand proportion; MBC: microbial
biomass carbon; BA: soil bacterial abundance; FA: soil fungal abundance. Values are
means ± standard error (SE). *p* values represent significant differences between the
topsoil and subsoil according to Tukey's test.



**Supporting information**

Additional supporting information may be found online in the supporting information

tab for this article.