# Peer review of "Biotic factors dominantly determine soil inorganic carbon stock across"

_EGUsphere, 2022_

## Author Comment (AC1)

**Answers to review comments one – RC1**

**Pan et al., "Biotic factors dominantly determine soil inorganic carbon stock across Tibetan alpine grasslands" (egusphere-2022-562)**

**Reviewers' comments:**

**Referee 1:**

Comments to the Author

I read your manuscript with great interest. In my opinion, this article provides interesting data and insights on SIC dynamics, and thus fits the SOIL journal aim and scope. The article is well structured and organized, and written in an acceptable English. The study is about the results and analysis of a large-scale soil sampling campaign in the Tibetan plateau, aimed at investigating the soil inorganic carbon density in topsoil (0-10 cm) and subsoil (20-30 cm) and their correlation with various explanatory variables selected by the Authors. As such, I think the study should be published in SOIL, after some fixing of the manuscript, especially of the materials and methods and result section for what concerns the analysis of explanatory variables for SIC density.

**Response**: Thank the reviewer very much for the positive and insightful comments on our manuscript "Biotic factors dominantly determine soil inorganic carbon stock across Tibetan alpine grasslands". We sincerely appreciate the constructive comments and encouragement on the manuscript. We have carefully studied these valuable comments and revised the manuscript accordingly. Please see our point-by-point responses below regarding all the concerns. We have revised and added some sentences in the section of materials and methods to make them clear (Lines 124-131, 158-162, 197-198, 202-207, 217-218, and 222-229), and the results of the analysis of explanatory variables for SIC density were also modified (Lines 254-256).

Major comments

I think that the introduction is quite clear, but could be improved by: (i) giving an idea of the relative relevance of SOC/SIC pools, just to put things in perspective for readers;

**Response**: Thank you for the valuable suggestions. We have added a section to highlight the relative relevance of SOC/SIC pools as follows: "To alleviate the elevated levels of atmospheric carbon dioxide, most previous studies concentrate on the SOC pool because it responds quickly to global climate change such as warming and nitrogen deposition, and it is strongly linked with various ecosystem functions (Wang et al., 2002; Yang et al., 2012). " (Lines 41-45).

Wang, G. X., Qian, J., Cheng, G. D., Lai, Y. M.: Soil organic carbon pool of grassland soils on the Qinghai-Tibetan Plateau and its global implication, Sci. Total. Environ., 291(1-3), 207-217, https://doi.org/10.1016/S0048-9697(01)01100-7, 2002.

Yang, Y. H., Ji, C. J., Ma, W. H., Wang, S. F., Wang, S. P., Han, W. X., ... Smith, P.: Significant soil acidification across northern China's grasslands during 1980s-2000s, Glob. Chang. Biol., 18(7), 2292-2300, https://doi.org/10.1111/j.13652486.2012.02694.x, 2012.

(ii) give some definition of "top" and "sub" soil to the reader (different researchers may divide the soil profile in different ways).

**Response**: Thank the reviewer very much for the critical comments. We agree with the comment that different definitions of topsoil and subsoil were reported in previous studies (Yost and Hartemink, 2020; Rumpel et al., 2012). Based on our field observation, the soil depth is relatively shallow (less than 40 cm) for alpine grasslands, especially for the alpine desert. Moreover, most of the belowground roots in alpine grasslands distribute on the surface of 10 cm and decrease sharply below 20 cm. Thus, we defined the topsoil and subsoil as 0-10 cm and 20-30 cm soils, respectively. We have added this information in the new manuscript (Lines 39-73 and 124-128).

Yost, J. L., & Hartemink A. E.: How deep is the soil studied – an analysis of four soil science journals, Plant Soil, 452, 5-18, https://doi.org/10.1007/s11104-020-04550-z, 2020.

Rumpel, C., Chabbi, A., Marschner, B. 2012.: Carbon storage and sequestration in

subsoil horizons: Knowledge, gaps and potentials, p 445–464. In Lal, R., Lorenz, K., Huttl. R. F., Uwe Schneider, B, von Braun, J. (ed), Recarbonization of the biosphere: ecosystems and the global carbon cycle. Springer, Heidelberg, Germany. https://doi.org/10.1007/978-94-007-4159-1_20.

The materials and methods section is mostly good, but I have a few comments/reservations: (i) did the Authors think that taking part of the samples in July, part in August, and part in September, may have had an effect on the results? For example, maybe soil pH and microbial abundance vary during summer, and thus there is another explanatory variable not taken into account (temporal variation). I suggest to include the information on date of sampling in Table S1, and discuss this issue in the Discussion section.

**Response**: Thank the reviewer very much for the critical comments. In practice, we have tried our best to shorten the time span of taking samples, which is labor-intensive. We have rechecked and added the information of sampling date sampling in Table S1. And the detailed sampling date was 30 days from 30 July to 28 August 2020 (Line 102), which was considered to have little effect on the results, due to the relatively stable plant growth stage and environmental conditions from July to August on the Tibetan Plateau.

(ii) The subsection "Statistical analyses" need improvement, in my opinion; more specifically, it needs to be more rigorous. First of all, a clear list of all the explanatory variables taken into account should be given, and a clear definition of which goes into edaphic, microbial, plant, and climate - and also biotic/abiotic.

**Response**: Good suggestions. We have added a list of all explanatory variables and defined the edaphic, microbial, plant, and climate - and also biotic/abiotic factors in the Supplement materials in Table S2 (Lines 217-218).

Table S2 Explanatory variables were categorized into the climatic, edaphic, plant, and microbial factors, and also abiotic/biotic factors.

| Variables | Abbreviations | Category | Factors |
|---|---|---|---|
| Soil inorganic carbon | SIC | - | - |
| Mean annual precipitation | MAP | Climatic factor | Abiotic factor |
| Mean annual temperature | MAT | Climatic factor | Abiotic factor |
| Altitude | - | Climatic factor | Abiotic factor |
| Plant aboveground biomass | PAB | Plant factor | Biotic factor |
| Plant coverage | PC | Plant factor | Biotic factor |
| Root biomass | RB | Plant factor | Biotic factor |
| pH | - | Edaphic factor | Abiotic factor |
| Soil available nitrogen | AN | Edaphic factor | Abiotic factor |
| Sand proportion | SP | Edaphic factor | Abiotic factor |
| Microbial biomass carbon | MBC | Microbial factor | Biotic factor |
| Bacterial gene abundance | BA | Microbial factor | Biotic factor |
| Fungal gene abundance | FA | Microbial factor | Biotic factor |

Then, there is a question: why didn't the Authors study the correlation index for each variable with respect to the target (SIC)? Spearman and Pearson correlations could be used, and give a clear picture to the reader in a simple table.

**Response**: We are very grateful to the helpful suggestions. We have added the Pearson correlation coefficients between SIC density and each variable for both the topsoil and subsoil in Table S3 in the revised manuscript (Lines 197-198). We found that the result of Pearson correlation was similar to that of linear regression.

Table S3. Pearson correlation coefficients between SIC density and explanatory variables. * and ** indicate significance at $p < 0.05$ and 0.01 (n=25), respectively.

| Parameter | MAP | MAT | Altitude | PAB | PC | RB | pH | AN | SP | MBC | BA | FA |
|---|---|---|---|---|---|---|---|---|---|---|---|---|
| Topsoil SIC | -0.59** | 0.07 | 0.43* | -0.56** | -0.67** | -0.41* | 0.70** | -0.56** | 0.54** | -0.43* | -0.50* | -0.52** |
| Subsoil SIC | -0.64** | -0.03 | 0.39 | -0.55** | -0.66** | -0.28 | 0.61** | -0.52** | 0.50* | 0.02 | -0.43 | -0.48 |

Then, instead of selecting the most relevant explanatory variables to build the multi-linear model, the Authors decide to create a large model with all explanatory variables; this I can understand, but the reason for this choice vs the former should be given.

**Response**: Thanks for the critical comments. We have added the reasons for selecting all explanatory variables as follows: "Each predictor variable was simultaneously tested in the model, which was comparable for the contribution of different types of predictors to SIC density. And the absolute values of standardized regression coefficients of the explanatory variables accounting for the percentage of the sum of all standardized regression coefficients were used to express the importance of predictors (Gross et al., 2017; Le Provost et al., 2020)" (Lines 202-207).

Gross, N., Le Bagousse-Pinguet, Y., Liancourt, P., Berdugo, M., Gotelli, N. J., Maestre, F. T.: Functional trait diversity maximizes ecosystem multifunctionality, Nat. Ecol. Evol., 1(5), https://doi.org/10.1038/s41559-017-0132, 2017.

Le Provost, G., Badenhausser, I., Le Bagousse-Pinguet, Y., Clough, Y., Henckel, L., Violle, C., Bretagnolle, V., Roncoroni, M., Manning, P., Gross. N.: Land-use history impacts functional diversity across multiple trophic groups, Proc. Natl. Acad. Sci. U. S. A., 117(3), 1573-1579, https://doi.org/ 10.1073/pnas.1910023117, 2020.

As far as I understand, the Authors create a "theoretical" multilinear model with all explanatory variables the Authors identified, and then the Authors assess the relevance of each explanatory variable. In my experience, this is often done using Global Sensitivity Analysis techniques, to take into account joint effects and different orders of sensitivity (see Saltelli 2008 sensitivity analysis a primer). However, the Authors use

another method, called Variation Partitioning analysis - that is okay with me, but this method should be explained further.

**Response**: Thank the reviewer very much for the critical comments. Variation partitioning analysis is a type of analysis that combines redundancy analysis (RDA) and partial RDA to divide the variation of a response variable among two, three or four explanatory data sets, which has been widely used in previous studies (Li et al., 2022; Qin et al., 2021; Yang et al., 2020). The results of variation partitioning analyses are traditionally represented by a Venn diagram, in which the percentage of explained variance by each explanatory data set is reported. We have added the explanation of the Variation Partitioning analysis as follows: "Furthermore, the relative importance of abiotic (climatic and edaphic) and biotic (plant and microbial) variables in predicting SIC density was quantified by performing variation partitioning analyses (VPA) (Borcard et al., 1992) and using the "vegan" package in R 4.1.3, which was used to divide the variation of SIC density among two types of explanatory variables for their individual and joint effects" (Lines 221-223).

Li, X. X., Huang, J., Qu, C. C., Chen, W. L., Chen, C. R., Cai, P., Huang, Q. Y.: Diverse regulations on the accumulation of fungal and bacterial necromass in cropland soils, Geoderma, 410, 115675, https://doi.org/10.1073/10.1016/j.geoderma.2021.115675, 2022.

Qin, S. Q., Kou, D., Mao, C., Chen, Y. L., Chen, L. Y., Yang, Y. H.: Temperature sensitivity of permafrost carbon release mediated by mineral and microbial properties, Sci. Adv., 7(32), eabe3596, https://doi.org/ 10.1126/sciadv.abe3596, 2021.

Yang, J. J., Wang, J., Li, A. Y., Li, G. H., Zhang, F.: Disturbance, carbon physicochemical structure, and soil microenvironment codetermine soil organic carbon stability in oilfields, Environ. Int., 135, 105390, https://doi.org/ 10.1016/j.envint.2019.105390, 2020

The results section has the same problems of the previous section: it needs to be more rigorous on the statistical part. Please avoid confusing statements as "positively

associated" and "negatively correlated" - if the Authors study correlation, then both are correlated, either positively or negatively. The significance of a correlation should be given (it is in the figures 2 and 3, but not in the text).

**Response**:  Thank the reviewer very much for the careful review.  We have carefully checked and revised our manuscript to avoid ambiguous writing (Lines 249, 250, 251, and 333). Following the comments, we have added the significance between SIC density and predictor variables (lines 253 and 345).

It is very important that statistical techniques are important to have a common ground, thus my focus on rigor, but they do not give clear-cut answers: the difference between topsoil and subsoil relevance of explanatory variables is not as big as it appears from the text of subsection 3.3, as it can be seen from the figures 2, 3 and 4.

**Response**:  We totally agree with the comments. The different statistical techniques e.g., T-test, linear regressions, linear-mixed effect model, and VPA were gradually adopted in our manuscript to validate the potential differences between topsoil and subsoil relevance of explanatory variables. As we can see that the explanatory variables vary with soil depths to predict SIC density. For instance, the absolute value of slope for the regression equation for each explanatory variable (Figs. 2 and 3) in different soil depths has large differences. We have added the detailed description as follows: "Also, the absolute value of slope for the regression equation for the most explanatory variables (except for AN, MAT, and MBC) in the topsoil was larger than that of the subsoil, especially for RB and SP (Figs. 2 and 3) " (Lines 254-256).

Figures 2 and 3 show that the SIC values are mainly clustered by "grassland type" (i.e. AM, AS, and AD), so much so that probably the best predictor would be just to consider the grassland type, something that could be done using Remote Sensing (not very informative about processes, I admit). This should at least be discussed in section 4.

**Response**: Thank the reviewer very much for the critical comments. We did not consider grassland type as a categorical variable for the analysis because the "grassland type" has already reflected the differences in climatic, plant, edaphic, and microbial

explanatory variables. We agree with the comments that some explanatory variables have an increase or decrease trend from the alpine meadow to the alpine desert. Thus, we have broadened the discussions in section 4 as follows:

"Also, the SIC density in both two soil depths appears to have an increase or decrease trend from the alpine meadow to the alpine steppe and alpine desert (Figs. 2 and 3). In the present study, for example, the alpine meadow has larger plant productivity than the alpine steppe, which implies that more plant above- and below-ground residues are deposited in alpine meadow soils compared to alpine steppe soils. Therefore, from the perspective of the whole ecosystem, the grassland type would be a better predictor for the quantity and distribution of SIC density. " (Lines 323-329)

Figure 5 is very confusing, since the PVA method was and not explained I cannot decript it myself, the caption is not very informative: how did the Authors get the percentages in the caption? What do the Authors mean with "unique effect" and "common interception"? What are the percentages in the figure? What do the Authors mean with "residuals"?

**Response**: We are sorry for our ambiguous writing and confusing interpretation of the original version. In the revision, we have added the explanation of VPA analyses and the interpretation of "unique effect", "common interception", "the percentages in the figure", and "residuals" in the part of Materials and Methods.

"Variation partitioning (Borcard et al., 1992) was used to divide the variation of SIC density among two types of explanatory variables for their individual and joint effects. In this analysis, the common and unique contribution of two sets of explanatory variables (abiotic and biotic variables) to the variation of SIC density are determined. And, the residuals are determined by a fraction of response variable variations, which can not be explained by any of the explanatory variables. The VPA method allows us to explore the variation clearly by the percentage of explanatory variables, which is easy to interpret and can be discussed in the context of SIC density."(Lines 223-229)

Borcard, D., Legendre, P., Drapeau, P.: Partialling out the spatial component of ecological variation, Ecology, 73 (3), 1045–1055, https://doi.org/10.2307/1940179, 1992.

The conclusion could be improved by directly answering the two questions at the end of introduction at the beginning, and clearly dividing results (of the statistical analysis) from interpretation.

**Response**: We sincerely appreciate the helpful comments. Following the comments, we have carefully revised the conclusions (Lines 407-408 and 414-416).

**Minor comments:**

**line - comment**

15 - start with "The"

**Response:** Done

23 - "associated" do the Authors mean "correlated"?

**Response:** Thank the reviewer very much for the careful review. We have revised it as follows "The SIC stock showed a significant increase with [...], but declined with [...]" (Lines 23-24).

51 - substitute "where" with "which"

**Response:** Corrected

84 - create a new paragraph for lines 84-91

**Response:** Thank the reviewer very much for the helpful suggestion. We have revised it accordingly (Line 91)

122 - "the rest of the samples, about 700 g, were also [...]"

**Response:** Done

123 - "other soil properties" which ones?

**Response:** We have added the relevant information (Lines 134-135).

127 - substitute "into" with "for"

**Response:** Done

128 - substitute "gravels" with "material"

**Response:** Done

166 - "[...] terminated. The primer [...]"

**Response:** Done

167 - substitute "Then" with "Finally"

**Response:** Done

219-220 - "(Fig.s 2 and 3 for topsoil and subsoil, respectively)"

**Response:** Thanks a lot. We have revised it.

220 - again "associated". Not clear what the Authors mean, and how it relates with "negatively correlated"

**Response:** Following the comments, we have revised the inappropriate words as follows: "the SIC density showed a significant increase with altitude, pH, and sand proportion, but declined with MAP, PAB, PC, RB, AN, BA, and FA (Lines 249)".

241 - substitute "afford" with "study"

**Response:** Corrected

244 - substitute "Since" with "Due to"

**Response:** Corrected

282 - substitute "more" with "larger"

**Response:** Corrected

293 - "association"?

**Response:** We have deleted the word "and negative" (Line 333)

304 - "[...] fungal gene abundance are correlated with SIC stock [...]"

**Response:** We have revised it following the suggestion (Lines 344-345)

308 - "and the increase in acidity is neutralized through [...]"

**Response:** We have revised it following the suggestion (Line 348)

350 - substitute "less roles" with "a lesser role"

**Response:** Done

355 - substitute "significance" with "effect"

**Response:** Done

356 - substitute "maintains" with "is important to maintain"

**Response:** Done

357-358 - "[...] biotic factors are correlated with SIC stock in the Tibetan plateau [...]"

**Response:** Done

**Thank you very much for your consideration.**

**Kind regards,**

**(Junxiao Pan and Jinsong Wang)**

---

## Author Comment (AC2)

**Answers to review comments one – RC2**

**Pan et al., "Biotic factors dominantly determine soil inorganic carbon stock across Tibetan alpine grasslands" (egusphere-2022-562)**

**Reviewers' comments:**

**Referee 2:**

Comments to the Author

Pan and co-authors studied how much inorganic carbon (SIC) is stored in soils throughout the Tibetan plateau along with biotic and abiotic parameters. In their main conclusion, they report a that biotic parameters excert greater control over SIC stocks than abiotic parameters, and that the significance of abiotic parameters is higher in the subsoil than in the topsoil.

**Response:** We thank the reviewer very much for summarizing our study and providing all the intellectual comments to further improve the manuscript.

The topic of this study – soil inorganic carbon stocks and their controls - is a timely and important, and the authors have collected an impressive dataset of biotic and abiotic measures. Their methods are state of the art and well described (although more details on plant parameter measurements are needed), and overall the manuscript is clearly written.

**Response:** We sincerely appreciate the reviewer for the positive comments and valuable suggestions to help us improve the manuscript. We have added the information on plant parameter measurements in the part of Material and Methods (Lines 158-162)

Unfortunately, I do not think that the second research question (contributions of different controlling factors to SIC stocks), on which the majority of the manuscript focuse, cannot be answered with the chosen experimental design, as correlation cannot proof causality. With this design, the authors can only show association of SIC stocks with external factors. In the most cases, it is not clear if e.g. FA, BA etc. influence SIC,

if SIC influences FA, BA, etc, or if both variables are independently influenced by an underlying third parameter.

In my opinion, this issue could be solved by rewriting large sections of the manuscript, removing wording like 'X has an effect on SIC', discussing potential controls in both directions as well as potential underlying third variables, and very, very carefully assessing if the partitioning into biotic and abiotic factors is still possible.

**Response:** We sincerely appreciate the helpful suggestions. And we agree with the comments that 'correlation cannot proof causation'. Following the intellectual comments, we have rewritten large sections of the manuscript and deleted some descriptions that are inappropriate. Specifically, we have removed the interpretation that overuses correlation as causative mechanism and deleted some descriptions (e.g., X has an effect on SIC, Lines 202, 212, 214, 220, 258, 263, 269-270, 296-298, 354, 364, 375, 387, 390, 400, 402-403, 409, 412 and 415) that are inappropriate and overusing correlation. Instead, we have also broadened the discussion of mechanisms underpinning our results by considering potential third variables as follows: "Although most of the variations in SIC density were explained by our measured explanatory variables, some other potential variables may also predict SIC density (Fig. 5). Then, understanding the effects of other potential abiotic and biotic factors on SIC density with soil depth is urgently needed when predicting the response and feedback of SIC to climate change in the future." (Lines 394-399)

One avenue for improving this manuscript would be to re-focusing it towards which parameters can be used to predict SIC content (rather than which parameters control SIC), which could be useful for mapping/upscaling of SIC stocks.

**Response**: We sincerely appreciate the reviewer for the constructive comments and helpful suggestions. We have carefully studied these useful comments and revised the manuscript accordingly. Specifically, we deleted the word " controlling" (Lines 202), or revised "controlling " into "predictor "(Lines 214 and 264). Also, we have revised some inaccurately expressions like 'X has an effect on SIC' into 'X parameter can be

used to predict SIC content' in the revised manuscript (Lines 202, 212, 214, 220, 258, 263, 269-270, 296-298, 354, 364, 375, 387, 390, 400, 402-403, 409, 412 and 415).

**Thank you very much for your consideration.**

**Kind regards,**

**(Junxiao Pan and Jinsong Wang)**